# Do Fire Cues Enhance Germination of Soil Seed Stores across an Ecotone of Wet Eucalypt Forest to Cool Temperate Rainforest in the Central Highlands of South-Eastern Australia?

Samuel Younis [†] [ID] and Sabine Kasel *[ID]

School of Ecosystem and Forest Sciences, Faculty of Science, The University of Melbourne, 500 Yarra Boulevard, Burnley, VIC 3121, Australia
* Correspondence: skasel@unimelb.edu.au
† Current address: Environs Kimberley, 44 Blackman Street, Broome, WA 6725, Australia.

**Abstract:** Soil seed banks play an important role in plant species persistence in fire-prone systems. Response to fire related germination cues often reflect historical fire regimes and can be important in maintaining ecotones between different forest types. We assessed the effects of heat and/or smoke on the soil stored seed banks across an ecotone of eucalypt to rainforest overstorey comprising wet forest, cool temperate mixed forest, and cool temperate rainforest in south-eastern Australia. Soils from five replicates of each forest type were subjected to very low (45 °C), low (65 °C) and high (90 °C) heat with or without two different smoke treatments: –smoke-infused vermiculite, and karrikinolide—a phytoreactive compound derived from smoke. Soils were placed in a glasshouse and germinants were identified and counted at weekly intervals. Response to fire cues was consistent among forest types despite underlying differences in the diversity of soil seed banks. There was no overall response of species richness, abundance, or composition to fire cues. Phanerophytes and ant-dispersed species with hard seed coats demonstrated positive response, and endozoochores negative response, to high heat independent of smoke. Endozoochores were concentrated (albeit at low densities) in cool temperate rainforest with no overall effect of seral affiliation on response to fire cues. Given the lack of response to karrikinolide, response to soil disturbance would most likely be associated with mechanical seed abrasion and/or exposure to increased light availability than to non-fire related production of smoke products. Forest type was a stronger determinant of floristics in the germinated soil seed bank than simulated fire related germination cues. Both smoke treatments had little influence on floristics in the germinated seed bank suggesting other, non-fire disturbances such as treefalls and soil turnover by fauna may be more important for germination for many of the species in these forest types.

**Keywords:** disturbance; *Eucalyptus regnans*; karrikinolide; *Nothofagus*; plant functional traits; smoke

---

## 1. Introduction

Fire is a pervasive and dominant disturbance of terrestrial ecosystems globally and plays a critical role in shaping plant communities [1]. The potential for increases in wildfire extent, frequency, and severity due to the removal of indigenous fire management [2] and climate change [3–5] is being realized at a global scale [6,7]. This shift in fire regime, particularly to more severe and frequent wildfire can threaten the persistence of plant species [8,9] and result in shifts in plant composition, including, in extreme cases, state changes, such as the conversion of forests to shrublands [10–12].

Fire severity and frequency drives forest succession in the Mountain Ash (*Eucalyptus regnans*) forests in the Central Highlands of south-eastern Australia. *Eucalyptus regnans* is an obligate seeding overstorey tree that is typically killed in high intensity fire, and regenerates from a canopy stored seedbank [10]. In contrast, the understorey regenerates from a soil seed store with succession following the initial floristics model of Egler [13–15]. The predominant

disturbance high-severity and infrequent stand-replacing fires that produce even-aged stands of *Eucalyptus regnans* [16]. Fires of low- to moderate-severity can produce multi-cohort stands, retaining biological legacies such as large trees [16,17]. Frequent, high-severity fire in sexually immature stands (<25 years) can result in regeneration failure and abrupt transition to acacia woodland via regeneration from a persistent soil seed store [10,17]. The long-term absence of fire (>200 years) may also result in forest transition, from eucalypt-dominated, wet sclerophyll forest; to cool temperate mixed forest—where eucalypt canopies support rainforest understories; and eventually cool temperate rainforest [13,16,18]. *Eucalyptus regnans* regeneration is dependent on the exposed mineral soils and high light conditions following fire [16,19]. Seedling development within the undistubed understorey is rare [16]. Maintenance of the eucalypt overstorey is therefore dependent on fire return intervals shorter than overstorey lifespan [20]. In contrast, rainforest species are shade tolerant and can germinate continuously under forest canopies and without the need for bare soil [21]. In mixed forest, rainforest species can continue to regenerate through gap dynamics. However, in the absence of fire, the eucalypt canopy cannot regenerate and will gradually die, with replacement by shorter rainforest trees [22–24]. Eucalypts can however seed into and return to recently burnt adjacent rainforest [22,25]. Fire regime is thus integral to the spatial and temporal distribution of wet eucalypt forest, mixed forest, and cool temperate rainforest in south-eastern Australia.

Cool temperate mixed forest often forms an ecotone between wet forest and rainforest and can provide a degree of protection to cool temperate rainforest (CTRF) from processes such as fire that maintain the adjacent wet forest [26]. Mixed forest is considered critical to the persistence of CTRF. The mix of species from both wet forest and CTRF can help slow and cool fires as they progress towards CTRF. Rainforest species support several traits that confer increased resistance and resilience to fire, including reduced leaf flammability and resprouting abilities [27]. For example, CTRF canopy species can resprout at high rates after fire [28] with Trouvé et al. [29] reporting 45% of *Nothofagus cunninghamii* and *Atherosperma moschatum* capable of resprouting following top kill by wildfire. Increased foliage cover from broad-leaved rainforest species alters fuel conditions and microclimate at ground level, providing an increasingly fire-suppressive environment [30–33]. As fire frequency and severity increases, CTRF becomes more open and can be invaded by light and fire tolerant species [24]. However, this invasion is more commonly observed in areas of mixed forest than CTRF. Both Chesterfield et al. [34] and Hill and Read [35] found that areas containing predominately rainforest species which can survive a patchy moderate fire will regenerate towards rainforest. Areas of mixed forest which have been burnt more severely and in closer proximity of a sclerophyll seed source will tend to regenerate towards sclerophyll dominated forest [35,36].

Fire-adapted traits including obligate seeding, resprouting, and fire-related germination cues are more in common in flammable than low-flammability forests where more plants regenerate independently of fire [37,38]. Soil seed bank response to fire-related germination cues often reflect historical fire regimes [39,40] and can be important in maintaining ecotones between different forest types [37,41]. For example, Tang et al. [41] demonstrated that fire-related germination cues, and in particular smoke, was important in maintaining an abrupt ecotone between subtropical rainforest and eucalypt wet sclerophyll forest in north-eastern Australia. Whether post-fire germination of species from the soil-stored seed bank will support the suite of existing CTRF species or cause a shift towards more light-demanding sclerophyllous wet forest species is a gap in our present understanding [42]. These different forest types are expected to have diverse responses to fire-related germination cues. Pausas and Lamont [40] suggest that fire-related germination cues are rare for non-fire prone ecosystems such as rainforest and where present, are confined to species shared with more fire prone habitats. Fabaceae species are common to wet forests, mixed forests and CTRF (e.g., acacias) and require heat shock to alleviate physical dormancy with germination triggered by soil temperatures which can exceed 80 °C [15,43,44]. These hard-seeded species are ant-dispersed with seed burial by ants acting to protect

seed from heat-induced mortality at the surface soil [40]. Conversely, some wet forest species, including shade tolerant species and those at eucalypt-rainforest boundaries can germinate in the absence of fire or may only germinate in response to low (<60 °C) soil temperatures [15,37,45].

Smoke has been shown to induce germination of a wide variety of species [46]. At the community level, the richness and abundance of seedlings germinating from soil seed banks are increased by smoke, particularly for crown fire ecosystems [40,47]. There is also evidence that smoke is an important germination cue for species in non-fire prone environments and that some rainforest species in Australia germinate in response to smoke [41,48–50]. Kasel et al. [15] suggest this mechanism is rare for eucalypt wet forest species following application of smoke-infused vermiculite in the absence of heat. Similarly, Carthey et al. [46] demonstrated species in wetter areas are less likely to germinate in response to smoke than those in drier habitats. To date, smoke response has been largely assessed using smoke water and aerosol smoke [46,51]. There has been very little research conducted on the response of soil seed banks to the active compound in smoke 'karrikinolide' [46] despite clear evidence that this active compound promotes greater rates of germination at extremely low concentrations with very little inhibitory effects at higher concentrations [52,53]. In contrast, traditional smoke products, which contain a mix of many different compounds, can inhibit the germination of some species [51,54]. These findings suggest that studies relying on aerosol smoke and smoke water may have underestimated the size and diversity of soil seed banks with previous work demonstrating that the abundance and diversity of seed bank germinants can be highly dependent on the type and concentration of the smoke product used [55].

Soil disturbance, for instance, following gap creation, may represent a non-fire related pathway for karrikinolide production through increased soil microbial oxidation of organic matter [52,56]. This premise was supported by several studies that have demonstrated that bacteria and fungi can generate karrikins [57–59]. Importantly, Ghebrehiwot et al. [60] measured considerable concentrations of karrikinolide in long unburnt (60 years) soils. Regeneration of rainforest species in response to gap creation suggest that physical disturbance is a key agent in germination and in turn that germination may be linked to non-fire related karrikinolide production [52].

This research aims to improve our understanding of fire-related germination cues of soil seed banks along a continuum of increasing moisture availability and time since last fire as characterized by transition from wet forest, to cool temperate mixed forest, and eventually cool temperate rainforest. We asked:

1.  Does the soil seed store respond to fire-related germinations cues and do these cues differ among the three forest types? We hypothesize that: (i) fire-related germination cues will be consistent across species shared among the three forest types; (ii) requirement of high-heat will be concentrated in ant-dispersed species with hard seed coats; and (iii) high heat, with or without smoke, will lead to a decline in rainforest-associated species.

2.  Does the application of karrikinolide increase germinant diversity and abundance and is this dependent on the application of heat? We hypothesize that: (i) any increase in germination to karrikinolide will be independent of heat; and (ii) rainforest-associated species will demonstrate a stronger response to karrikinolide than species associated with more flammable forest types.

3.  Does the abundance and diversity of soil seed banks differ among forest types? We hypothesize that wet forest and mixed forest will support larger and more diverse soil seed banks than rainforest and in particular, a greater proportion of short-lived, light demanding species more typical of earlier successional stages.

4.  Does the soil seed bank reflect species diversity in the extant vegetation and does this differ among the three forest types? We hypothesize that rainforest-associated species will be concentrated in extant vegetation and poorly represented in the soil seed bank, while the soil seed bank will support a greater abundance of early successional species.

## 2. Materials and Methods

### 2.1. Study Area and Site Selection

The study was conducted in the Central Highlands of Victoria, south-east Australia, approximately 100 km north-east of the city of Melbourne, Victoria (Figure S1). Sampling was restricted to three spatially contiguous stand types that dominate the regions forests: cool temperate rainforest (CTRF); wet sclerophyll eucalypt forest ('wet forest', WF); and an ecotone described as cool temperate mixed forest ('mixed forest', MF) where rainforest and wet forest species mix [18]. Rainforest stands are dominated by obligate rainforest species in all strata, including *Nothofagus cunninghamii* overstorey 30–40 m tall, the occasional emergent *Eucalyptus regnans* to 60 m tall, and understorey of *Atherosperma moschatum* to 20 m tall. *Pittosporum bicolor* and tree ferns (*Cyathea australis*, *Dicksonia antarctica*) dominate the lower strata (5–10 m) with ground ferns less than 1 m. Stands of wet forest are dominated by *Eucalyptus regnans* up to 80 m tall, above an understorey (20–40 m) of broad-leaved shrubs including *Acacia* spp., *Pomaderris aspera*, *Prostanthera lasianthos*, *Zieria arborescens* and *Persoonia arborea*. Mixed forest is a seral community where recurring disturbances prevent secondary succession to eucalypt-free CTRF and support an overstorey of eucalypts co-dominant with rainforest trees, above an understorey of rainforest species [18,20,61]. Despite being regarded as a seral community, mixed forest is considered sufficiently stable to persist as a distinct vegetation community in the landscape [62,63]. Both CTRF and mixed forest are listed as threatened ecological communities under the *Flora and Fauna Guarantee Act 1999*.

Five replicate sites were sampled for each forest type. Study sites encompassed flat to moderate slopes (1–14°) across a moderate elevation gradient (508–977 m asl) with annual rainfall from 1580 to 1730 mm and mean annual temperatures from 9.6 to 11.6 °C. Stand structural and edaphic conditions were originally surveyed by Fedrigo et al. [26] and varied significantly across the three forest types (Table S1). The most recent large-scale disturbance to have affected all sites was a broad landscape scale wildfire in 1939 [26].

### 2.2. Vegetation Survey

A 20 m × 20 m plot was established within each of the three forest types within each location (*n* = 15 plots, Figure S1). Vascular species in the above ground vegetation were recorded for each plot as presence-absence. Soil seed banks were estimated using the seedling emergence method. For this, the litter was gently removed and 40 soil cores to 5 cm depth (diameter 6.8 cm) [64,65] were randomly sampled in each quadrant and bulked for a total of 160 samples per plot. This sampling intensity is equivalent to a surface area of 0.581 m² and equivalent to or greater than those used elsewhere [15,66]. All field work was completed during the one sampling season from September to October 2020.

### 2.3. Soil Seed Bank Treatment

Soils were laid out for air-drying at room temperature immediately on return from the field. Once soil was dry it was sieved though an 8 mm × 8 mm sieve to break up large clumps of soil and remove rocks and organic debris, concentrating the seed in a smaller volume of soil [67]. For each site, the soil was weighed, once sieved, and equally divided into 10 parts for application of heat and/or smoke treatments (Table 1).

Heat promotes the germination of hard seeded species with physical dormancy such as in the Fabaceae [43] while for other species smoke, either in isolation or in combination with heat can increase seedling germination and richness [44,46]. For heat treatments, samples were placed in aluminum trays to a depth of 2 cm and heated in ovens to achieve a set maximum temperature range for a duration of 15 min (Table 1). Temperatures were measured at 1-cm depth in each heat treatment using thermocouples that logged temperatures at 1-min intervals. The HH reached 90 °C, the LH 65 °C and VLH 45 °C. Samples for each heat treatment were heated in the one oven to ensure there was no variation in heat across any given heat treatment (e.g., HH v HHS v HHK). Once cool, the samples were spread over 2 cm of sterilized pine bark in 29.5 cm × 35 cm seed raising

trays. Samples that were not heated (C, S, K, Table 1) were spread over the pine bark in the same way.

**Table 1.** Soil seed bank treatments.

| Abbreviation | Description | Maximum Temperature (°C) |
|---|---|---|
| C | Control | |
| S | Smoke (vermiculite product) | |
| K | karrikinolide | |
| LHS | Low heat plus smoke | 60–65 |
| HH | High heat | 85–90 |
| HHS | High heat plus smoke | 85–90 |
| HHK | High heat plus karrikinolide | 85–90 |
| VLH | Very low heat | 40–45 |
| VLHS | Very low heat plus smoke | 40–45 |
| VLHK | Very low heat plus karrikinolide | 40–45 |

Two treatments were used to simulate smoke-related germination cues. The smoke treatment (S) consisted of vermiculite infused with concentrated smoke extract which is derived from the burning of *Eucalyptus* and *Pinus* waste sawdust and percolating this smoke through water (Regen 2000, Grayson Trading, Australia). This smoke product was applied at the recommended concentration of 120 g m$^{-2}$. The karrikinolide, KAR1 (C8H6O3, 3-Methyl 2H-Furo[2,3-c]pyran-2-one) smoke treatment (K) was applied at a concentration of 0.67 μM at an application rate of 250 mL m$^{-2}$ [54]. The karrikinolide solution was applied to moist soil using a spray bottle with the first application on entry to the glasshouse and a second application applied a month later. KAR1 is a phytoreactive compound that is one of the main germination cues in smoke [52,53]. Trays with sterile pine bark media were used to monitor for contamination with the media subject to the same treatments as the seed bank material. Trays (15 sites × 10 treatments plus 10 controls) were placed into a climate-controlled glasshouse (~16 °C night, 22 °C day) with twice daily automatic irrigation.

Emerging seedlings were recorded at weekly intervals and removed from trays once identified. Unidentified seedlings were removed and grown in larger pots until they were identified. After 30 weeks there was little further germination and soils from all treatments were then turned to promote germination of any remaining seeds. Seedling trays were monitored for seedling emergence for a further 7 weeks. Soils were not examined for seed that had not germinated.

*2.4. Functional Types*

Functional types associated with resource partitioning and response to disturbance (life form, fire response, seral stage) and the capacity to re-colonize (dispersal mode) were recorded for all species [68]. Life form classification followed Raunkiaer [69] and classification of attributes for dispersal mode was based on diaspore morphology [68] (Table 2).

**Table 2.** Description of plant traits and their defining attributes.

| Trait and Attributes | Abbreviation | Description |
|---|---|---|
| Life form [A] | | |
| Chamaephyte | C | Persistent buds ≥1 cm and <20–30 cm above ground surface |
| Geophyte | G | Persistent buds buried to a depth of 2–3 cm |
| Hemicryptophytes | H | Persistent buds are in the immediate vicinity of the soil surface only, maximum height 1 cm |
| Phanerophyte | P | Persistent buds >20–30 cm on stems above the ground, includes twiners, vines, and epiphytes |
| Therophyte | T | Annual (monocarpic) plants, includes some facultatively perennial plants (polycarpic) that were judged to be predominantly annual |
| Dispersal mode [B] | | |
| Anemochory | ane | Wind-dispersed; pappus, coma, samara, or similar attachment |
| Barochory | bar | Gravity-dispersed; no apparent seed dispersal mechanism |
| Endozoochory | end | Ingestion by vertebrates (mainly mammals and birds); fleshy dispersal units (berries, drupes, or aggregate fruits) |
| Epizoochory | epi | Dispersal by adhesion to the outside of animals—usually on the hair of mammals, via appendages including barbs, hooks, spines, burrs, or awns |
| Mobile | mob | Long-distance wind dispersal of small seed; includes barochores with seed dimensions <0.5 mm, and mass generally <0.1 mg |
| Myrmecochory | myr | Ant-dispersed, elaiosome attached to seed to attract ants |
| Fire Response [C] | | |
| Obligate resprouters | R | Plants that rely on resprouting to regenerate after fire |
| Obligate seeders | S | Plants that do not resprout and rely on seeding to regenerate after fire |
| Facultative seeders | SR | Plants that can resprout and germinate seeds after fire |
| Weak seeders | Sr | Seeders that also have some capacity to resprout |
| Weak resprouters | Rs | Resprouters that have some capacity to regenerate from seed |
| Seral Stage [D] | | |
| Renewal to founding (early) | early | <3 years post fire |
| Juvenile | juv | 3–9 years post fire |
| Adolescent | adol | 9–35 years post fire |
| Mature | mat | 35–250 years post fire |
| Waning to Senescence (rainforest) | rf | ≥250 years post fire (rainforest associated species) |
| Generalist | gen | Species occurring across all seral stages <250 years |

[A] [69]; [B] [68]; [C] [70]; [D] [15].

Fire response categories follow the Australian Fire Ecology Database [70] and seral stage follow Kasel et al. [15]. In subsequent analyses, species were excluded where attribute information was missing or where attributes were represented by fewer than three species (Table S2).

*2.5. Statistical Analyses*

Glasshouse contaminants recorded in the seed bank germination trail (*Epilobium* spp., *Cardamine occulata*) were removed from the dataset prior to analysis.

2.5.1. Fire-Related Germination Cues and Forest Type

We calculated the number of seedlings that germinated in each of the ten treatments across three levels of data aggregation: (i) all species; (ii) functional types; and (iii) individual species. For functional types, species richness and germinant density was summed for species within each life form, dispersal mode, fire response strategy and seral stage.

Non-parametric Permutational Multivariate Analysis of Variance (PERMANOVA Version 1.0.5, PRIMER-E Ltd., Plymouth, UK; [71,72]) was used to test effects of seed bank treatment (10 factors) and forest type (three factors: CTRF, MF, WF) on species richness (all species, species within each functional type), germinant abundance (total abundance, abundance within each functional type) and species diversity (H'), using the Euclidean distance matrix. Diversity was calculated using Shannon's diversity index using the formula:

$$-SUM(Pi*ln(Pi)) \tag{1}$$

where Pi is the proportion (*n*/N) of individuals of one particular species (i) found (*n*) divided by the total number of individual species found (N). Where needed, richness

and abundance were $\log_{10}(x + 1)$ transformed prior to analysis to meet assumptions of normality.

PERMANOVA also tested for effects of seed bank treatment and forest type on species composition (Bray-Curtis similarity); and functional trait associations (Euclidean distance). In each instance, data were standardized (division by the total) and transformed ($\log_{10}(x + 1)$) counts of each species [39,73,74]. For the trait associations, counts are totals across species within each life form, dispersal mode, fire response strategy, and seral affiliation. SIMPER was used to determine the species or traits that contributed most to dissimilarity.

For individual species, significance of treatment and forest type effects were limited to those species recorded in ten or more sub-samples and only included sites in which each species had germinated [75]. Data were standardized (division by the total) and transformed ($\log_{10}(x + 1)$) prior to analysis. Assumptions of normality could not be met, and analysis followed Kruskal-Wallis and Dunn's pairwise comparisons (where $p \leq 0.05$; [76]). Kolmogorov-Smirnov test was used to examine significance of differences in the time-courses of seed germination among treatments [76].

2.5.2. Associations between Vegetation Pool and Forest Type

Given that each soil seed bank treatment produced a unique set of additional species, comparison with extant vegetation was based on soil seed bank germinants summed across all treatments as this approach provides for the most comprehensive account of the germinable soil seed bank (e.g., [77]). All species in the extant vegetation and those that had germinated in the soil seed bank were considered.

For each of two levels of data aggregation (all species, functional types—as above), two-way PERMANOVA as used to assess vegetation pool (two factors: extant vegetation, soil seed bank) and forest type (three factors: CTRF, MF, WF) effects on species composition and trait associations. Analysis was based on the Bray-Curtis similarity matrix using presence–absence transformed data of species composition and on Euclidean distance of square-root transformed counts of the frequency distribution of traits [78]. SIMPER was used to determine the species or traits that contributed most to dissimilarity.

Significance of differences in species richness (total species, native species, introduced species) was based on two-way PERMANOVA on Euclidean distance between sites.

## 3. Results

### 3.1. Fire-Related Germination Cues

A total of 5028 germinants were recorded in the soil seed bank, comprising 75 species from 27 families (Table S2). There were 11 introduced species that comprised just 1.4% of the total germinant pool. *Acacia dealbata* and *Hydrocotyle hirta* were the most abundant species, while 39 species were represented by fewer than 10 germinants. Barochores (39% of total germinants across all dispersal modes), myrmecochores (35%), hemicryptophytes (47% across all life forms), phanerophytes (49%) and generalists (69% across all seral types) dominated the soil seed store with few differences in the relative proportion of fire response attributes (Figure S2). There were 22 species that were unique to a specific soil seed bank treatment, however each of these had fewer than 5 germinants except *Veronica calycina* with 50 germinants restricted to HHK. There were 23 species limited to one forest type, all with fewer than 5 germinants except for *Cyperus lucidus* with 126 germinants and restricted to CTRF. *Nothofagus cunninghamii* was the one other species limited to CTRF with just one germinant recorded in the soil seed bank (Table S2).

The frequency distributions for seedling emergence in all treatments were not significantly different from each other (Kolmogorov-Smirnov two-sample test, $p > 0.05$; Figure S3).

### 3.2. Individual Species

For the 23 species for which statistical testing was possible, six species demonstrated significant differences in seedling densities in response to germination treatments (Table S2).

The application of K increased germinant density of *Lepidosperma elatius* above S and C, with similar response in combination with VLH treatments. In contrast germinant density of *Viola hederacea* declined with K relative to S and C with no differences for smoke treatments in combination with heat. The germinant density of *Acacia dealbata* increased with heat independent of smoke treatment (either S or K). In contrast, high heat, either independently or in combination with smoke treatment produced lower density relative to control in *Australina pusilla* subsp. *muelleri* (Table S2). Ten species demonstrated significant forest type effects including *Cassinia aculeata*, *Olearia phlogopappa* subsp. *continentalis*, *Stellaria flaccida* and *Tetrarrhena juncea* that were more abundant in WF than MF and CTRF. CTRF supported greater abundance of *Isolepis inundata* and *Juncus procerus* than MF or WF (Table S2).

### 3.3. All Species

Species richness, abundance, diversity (H) and composition differed among forest types but not treatment, with no interaction effects (Table S3, Figure 1). Species richness, abundance, and diversity in WF was significantly greater than CTRF and MF, with no differences between CTRF and MF (Figure 1). Species composition differed among all forest types. *Acacia melanoxylon*, *Zieria arborescens* and *Pomaderris aspera* made the greatest contributions to differences among forest types with A. melanoxylon most strongly associated with CTRF, and *Z. arborescens* and *P. aspera* with WF (SIMPER, Table S4).

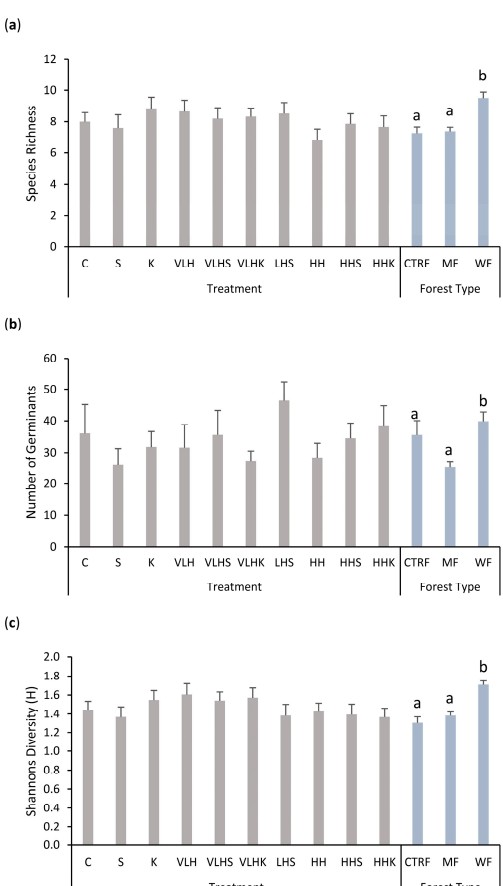

**Figure 1.** (**a**) Species richness, (**b**) abundance, and (**c**) Shannon's Diversity (H) of soil seed bank germinants by treatment and forest type. Values are site means and bars one standard error. There were no significant differences among treatments. For forest type, letters indicate significant ($p \leq 0.05$) pairwise differences. Statistical analysis was based on $\log_{10}(x + 1)$ transformed values of abundance, but graphical display generated from raw data. Model results provided in Table S3. There were no significant interaction effects.

### 3.4. Functional Traits

The abundance and richness of attributes differed among soil seed bank treatments and forest types, with response to treatments consistent across forest types (i.e., no significant interaction; Table S5). The abundance of phanerophytes, therophytes, myrmecochores, and endozoochores differed among treatments (Figure 2). Myrmecochore and phanerophyte abundance increased with increasing heat, but there was no increase in response to S and K (Figure 2). Endozoochore abundance and richness declined with heat, with however no differences among smoke treatments. Therophyte response to germination treatment was limited to HHK where abundance exceeded HH, due to the positive response of *Juncus bufonius* (Table S2). The abundance and richness of anemochores, barochores, and obligate seeders declined from WF to MF to CTRF. The abundance and richness of phanerophytes and early successional species was greater in WF than MF and CTRF. In contrast CTRF supported more endozoochores, mobile species, resprouters, and rainforest associated species than MF (Figure S4).

There were no significant differences in functional trait associations among soil seed bank treatments (*Pseudo-F* = 1.0, *p* = 0.3744). Trait associations did however differ among forest types (*Pseudo-F* = 7.8, *p* = 0.0001) with no significant interactions (Table S6). Trait associations differed among all forest types with increased abundance of rainforest seral affiliation, mobile and obligate resprouters in CTRF relative to MF and WF, and greater abundance of anemochores in WF relative to other forest types (SIMPER, Table S7, Figure S5).

### 3.5. Vegetation Pool Versus Forest Type

Species richness, species composition, and functional trait associations differed significantly between vegetation pool and forest type with no significant interaction effects.

#### 3.5.1. Soil Seed Bank Versus Extant Vegetation

Across both vegetation pools, 43% of species were unique to the soil seed bank, 30% of species were restricted to extant vegetation and 27% of species were common to both pools. The soil seed store supported a significantly greater number of total and introduced species than the extant vegetation (Table 3). The frequencies of species with different life forms, dispersal mechanisms, fire response strategies and seral affiliation (Figure 3) varied significantly between vegetation pools (Table 3). The soil seed bank supported more hemicryptophytes, therophytes, myrmecochores, epizoochores, and early successional species; while extant vegetation supported more obligate resprouters and rainforest associated species with these attributes contributing most to dissimilarity between vegetation pools (Table 3).

**Table 3.** Mean number of species per site (across all forest types) with each attribute in the soil seed bank and extant vegetation and the mean contribution to dissimilarity between vegetation pools as determined by SIMPER tests (following PERMANOVA, *Pseudo-F* and *p* values shown). Attributes are ordered according to their contribution to dissimilarity.

| Trait and Attribute | Soil Seed Bank | | Extant Vegetation | | Dissimilarity (%) |
|---|---|---|---|---|---|
| | mean | SE | mean | SE | |
| Life form | *Pseudo-F* = 35.3, *p* = 0.0001 | | | | |
| Hemicryptophyte | 9.80 | (0.82) | 3.80 | (0.62) | 36.9 |
| Therophyte | 2.20 | (0.40) | 0.20 | (0.11) | 33.6 |
| Phanerophyte | 8.07 | (0.57) | 11.07 | (0.69) | 10.4 |
| Geophyte | 1.07 | (0.07) | 3.00 | (0.24) | 10.0 |
| Chamaephyte | 1.53 | (0.17) | 0.93 | (0.21) | 9.1 |
| Dispersal mode | *Pseudo-F* = 9.7, *p* = 0.0001 | | | | |
| Myrmecochore | 4.67 | (0.35) | 2.07 | (0.34) | 24.9 |
| Epizoochore | 1.27 | (0.21) | 0.47 | (0.17) | 21.1 |
| Barochore | 6.53 | (0.47) | 4.47 | (0.72) | 19.4 |

**Table 3.** *Cont.*

| Trait and Attribute | Soil Seed Bank | | Extant Vegetation | | Dissimilarity (%) |
|---|---|---|---|---|---|
| Endozoochore | 3.07 | (0.46) | 3.07 | (0.30) | 13.0 |
| Anemochore | 4.07 | (0.68) | 3.40 | (0.19) | 10.8 |
| Mobile | 3.60 | (0.29) | 5.87 | (0.35) | 10.7 |
| Fire response | | *Pseudo-F* = 5.1, *p* = 0.0002 | | | |
| Facultative seeder (SR) | 1.27 | (0.27) | 1.47 | (0.13) | 38.4 |
| Obligate resprouter (R) | 6.40 | (0.70) | 8.00 | (0.65) | 29.4 |
| Obligate seeder (S) | 6.73 | (0.65) | 4.60 | (0.36) | 20.8 |
| Weak seeder (Sr) | 2.27 | (0.21) | 1.73 | (0.27) | 6.9 |
| Weak resprouter (Rs) | 4.40 | (0.24) | 3.07 | (0.25) | 4.4 |
| Seral stage | | *Pseudo-F* = 42.1, *p* = 0.0001 | | | |
| Early | 7.53 | (1.00) | 1.80 | (0.58) | 53.5 |
| Rainforest | 1.00 | (0.14) | 6.00 | (0.61) | 33.3 |
| Generalist | 12.47 | (0.82) | 10.27 | (1.23) | 13.2 |
| Origin | | | | | |
| | | *Pseudo-F* = 1.3, *p* = 0.256 | | | |
| Native | 20.00 | (1.06) | 18.53 | (1.09) | |
| | | *Pseudo-F* = 25.1, *p* = 0.0002 | | | |
| Introduced | 1.80 | (0.30) | 0.47 | (0.17) | |
| | | *Pseudo-F* = 12.5, *p* = 0.0021 | | | |
| All species | 24.40 | (1.29) | 19.33 | (1.26) | |

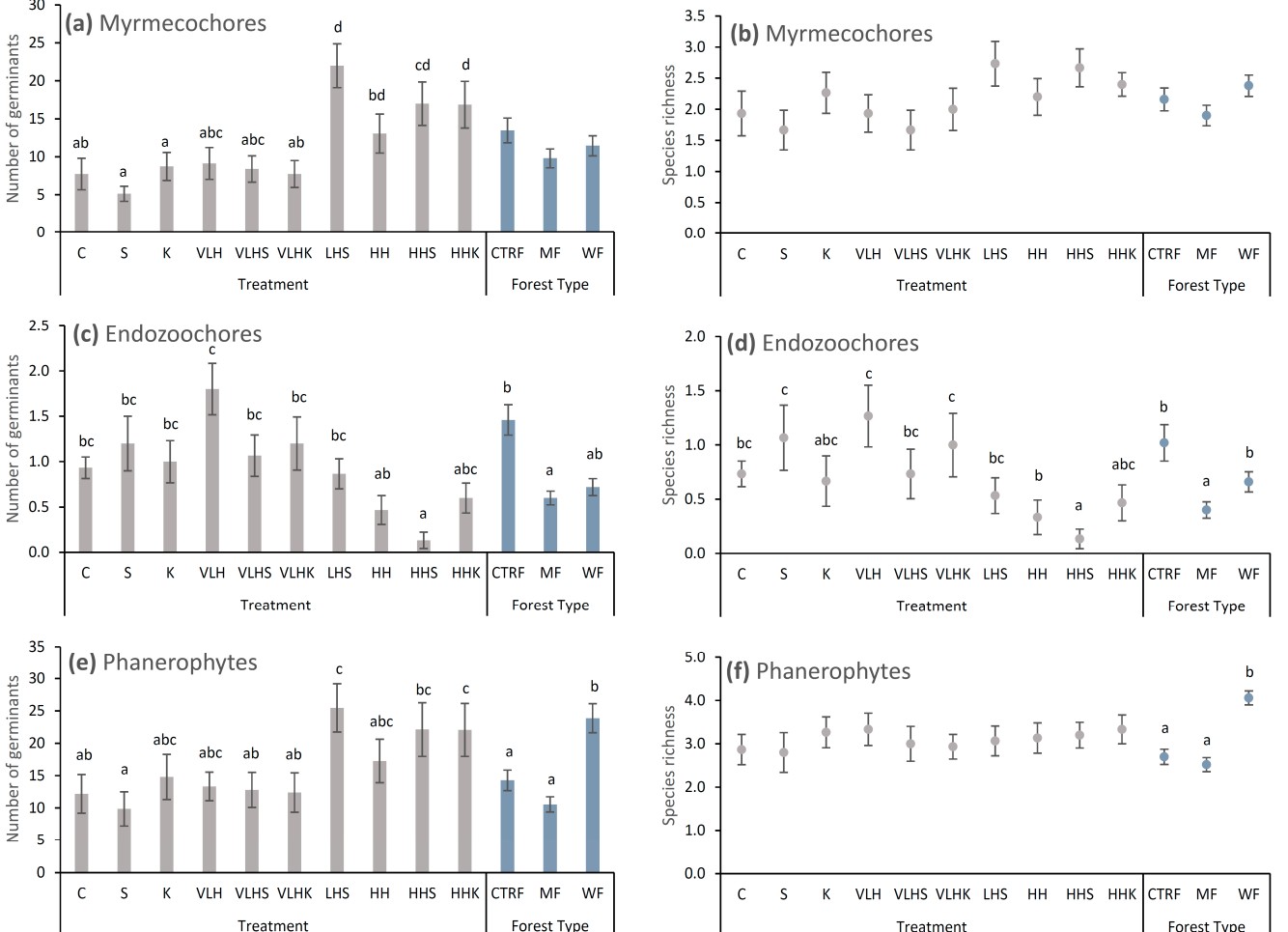

**Figure 2.** *Cont.*

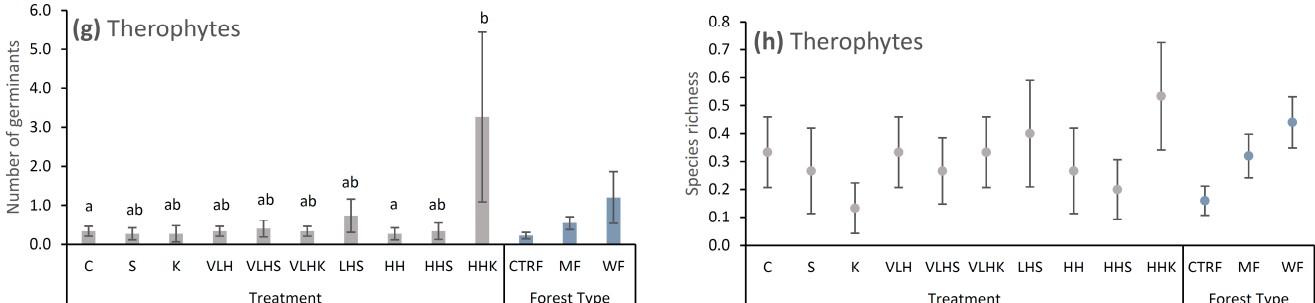

**Figure 2.** Soil seed bank treatment (see Table 1) and forest type (CTRF, cool temperate rainforest; MF, mixed forest; WF, wet forest) effects on germinant density (bars) and richness (circles) (mean per site, standard error) of: (**a,b**) myrmecochores, (**c,d**) endozoochores, (**e,f**) phanerophytes, and (**g,h**) therophytes. Significant pairwise comparisons ($p \leq 0.05$) within treatments or forest type are indicated by different letters. There were no significant interaction effects. Soil seed bank treatment effects were not significant for other functional attributes with model results provided in Table S5.

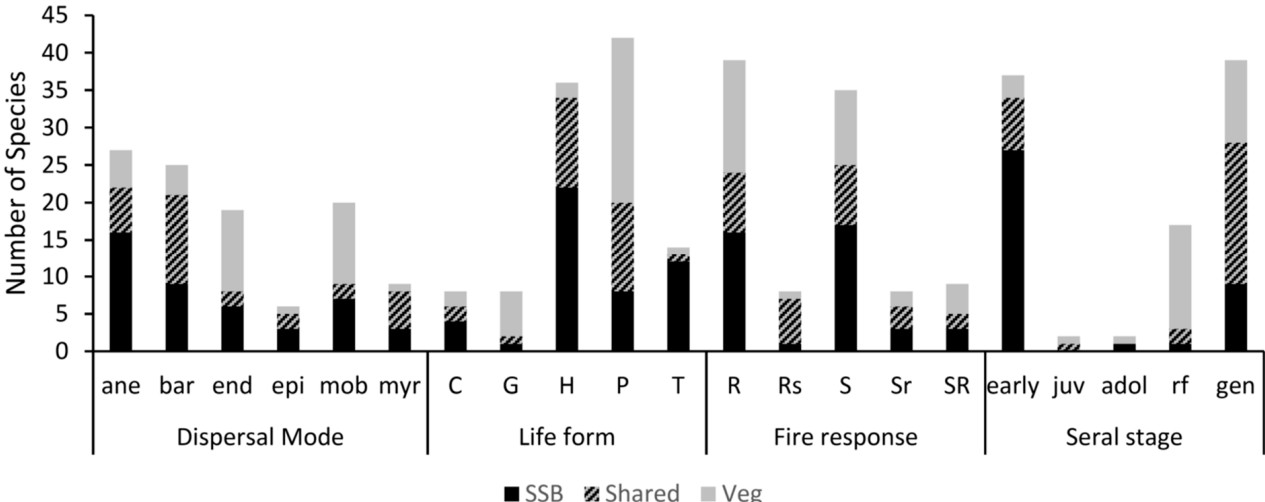

**Figure 3.** Frequency of plant functional types restricted to the soil seed bank (SSB) or extant vegetation (Veg), and those shared between vegetation pools. Dispersal mode: ane, anemochory; bar, barochory; end, endozoochory; epi, epizoochory; mob, mobile; myr, myrmecochory. Life form: T, therophyte; H, hemicryptophyte; C, chamaephyte; P, phanerophyte; G, geophyte. Fire response: R, obligate resprouters; Rs, weak resprouters; S, obligate seeders; SR, facultative seeders; Sr, weak seeders. Seral stage: juv, juvenile; adol, adolescent; gen, generalist; rf, rainforest. See Table 2 for definitions of functional types. Data were excluded where trait information was missing (see Table S2).

Mean pairwise similarity in composition between extant vegetation and the soil seed bank was low (28%) and PERMANOVA indicated a significant (*Pseudo-F* = 19.9, $p < 0.0001$) turnover between vegetation pools that is reflected in the clear separation of sites in the NMDS ordination (Figure 4). SIMPER indicated that species most responsible for the differences between vegetation pools included *Blechnum wattsii*, *Dicksonia antarctica*, *Nothofagus cuninghamii* and *Eucalyptus regnans* which were more frequent in the extant vegetation, and *Acacia* spp., *Gahnia* spp. and *Cassinia aculeata* that were more frequent in the soil seed bank (see Figure 4, Tables S8 and S9).

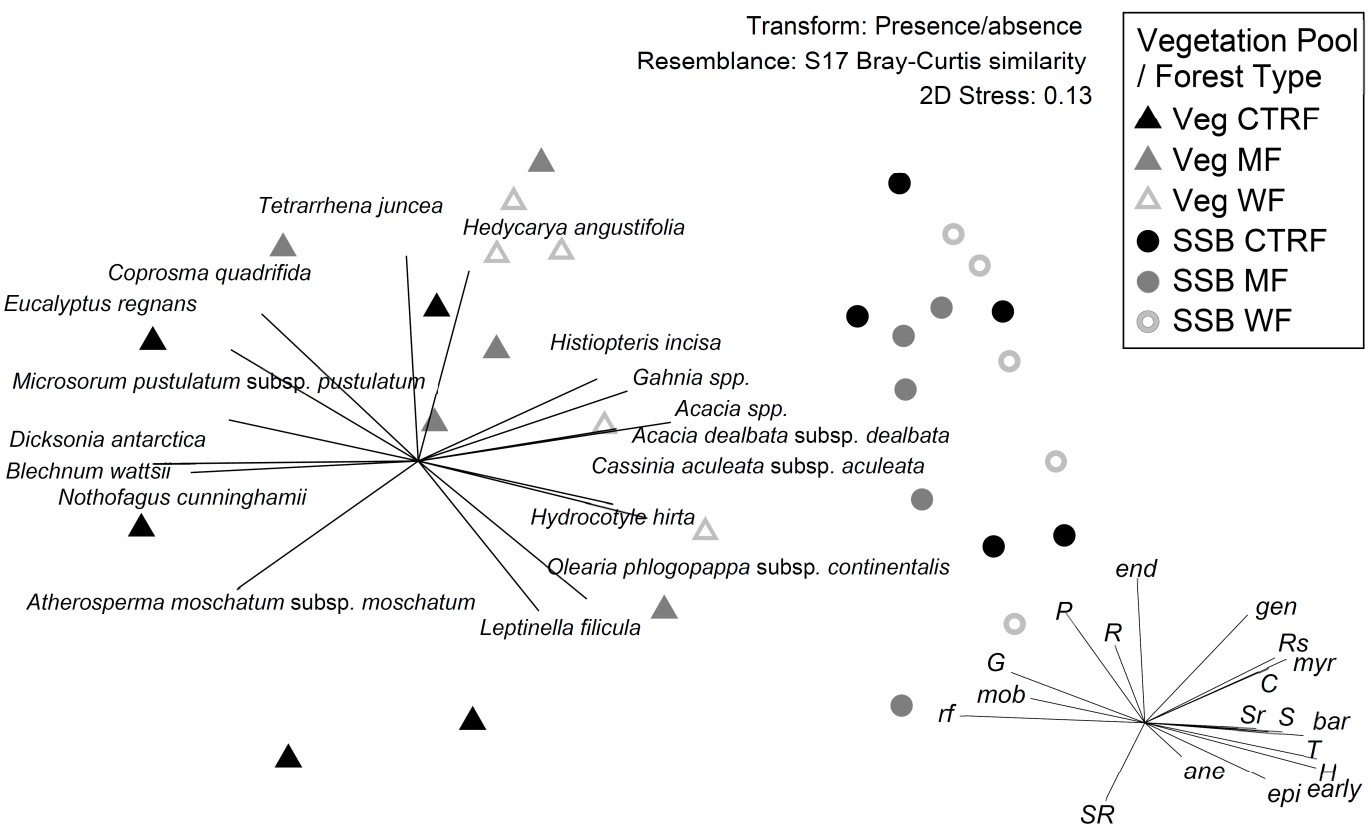

**Figure 4.** NMDS ordination of presence–absence data for species in the germinable soil seed bank (SSB) and extant vegetation (Veg) across three forest types (CTRF, cool temperate rainforest; MF, mixed forest; WF, wet forest) in the Central Highlands of south-eastern Australia. Symbols represent individual sites and are coded according to vegetation pool (shape) and forest type (colour). The vector overlay represents the strength and direction of the relationship for underlying species with $r \geq 0.65$. The vector insert (bottom right) represents relationships with all underlying plant functional types. Life form: C, chamaephyte; G, geophyte; H, hemicryptophyte; P, phanerophyte; T, therophyte. Dispersal mode: ane, anemochory; bar, barochory; end, endozoochory; epi, epizoochory; mob, mobile; myr, myrmecochory. Fire response: R, obligate resprouters; S, obligate seeders; SR, facultative seeders; Sr, weak seeders; Rs, weak resprouters. Seral stage: gen, generalist; rf, rainforest.

3.5.2. Forest Type

Each forest type supported a unique set of species across the combined soil seed bank and extant vegetation pools of diversity (Figure 5). The mixed and wet forests supported a greater number of species than CTRF (Table 4). The frequencies of species with different life forms, dispersal mechanisms, fire response strategies, and seral affiliation varied significantly between forest types (Table 4). The mixed and wet forests supported more hemicryptophytes, therophytes, and early successional species than CTRF; while wet forest supported more myrmecochores, epizoochores, and obligate seeders than CTRF with these attributes contributing most to dissimilarity between vegetation pools (Table 4).

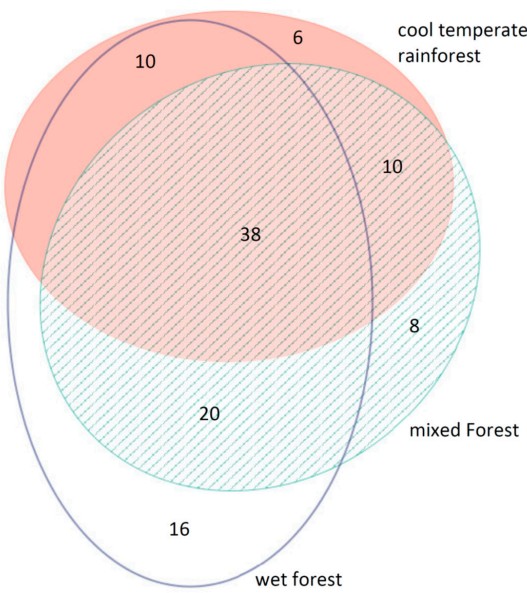

**Figure 5.** Venn diagram showing the number of unique and shared species among the three forest types. Numbers represent the combined pool of species in extant vegetation and soil seed bank. Solid color = cool temperate rainforest; Diagonal fill = mixed forest; No fill = wet forest.

**Table 4.** Mean number of species per site (across both vegetation pools) with each attribute per forest type and the mean contribution to dissimilarity between forest types as determined by SIMPER tests (following PERMANOVA, *Pseudo-F* and *p* values shown). Attributes are ordered according to their contribution to dissimilarity. Letters indicate significant pairwise differences ($p \leq 0.05$). CTRF, cool temperate rainforest.

| Trait and Attribute | CTRF | | Mixed Forest (MF) | | Wet Forest (WF) | | Dissimilarity (%) | |
|---|---|---|---|---|---|---|---|---|
| | mean | SE | mean | SE | mean | SE | | |
| Life form | *Pseudo-F* = 3.2, *p* = 0.0043 | | | | | | | |
| | a | | b | | b | | CTRF v WF | CTRF v MF |
| Hemicryptophyte | 5.9 | (1.5) | 7.1 | (1.2) | 7.4 | (1.3) | 33.6 | 38.8 |
| Therophyte | 0.7 | (0.3) | 1.1 | (0.4) | 1.8 | (0.7) | 31.1 | 23.4 |
| Chamaephyte | 0.7 | (0.2) | 1.5 | (0.2) | 1.5 | (0.2) | 15.4 | 17.6 |
| Phanerophyte | 8.4 | (0.9) | 9.2 | (0.9) | 11.1 | (0.7) | 11.7 | 12.9 |
| Geophyte | 1.6 | (0.3) | 2.1 | (0.4) | 2.4 | (0.5) | 8.1 | 7.3 |
| Dispersal mode | | | *Pseudo-F* = 2.4, *p* = 0.0132 | | | | | |
| | a | | ab | | b | | CTRF v WF | |
| Myrmecochore | 3.2 | (0.7) | 3.2 | (0.6) | 3.7 | (0.5) | 22.5 | |
| Epizoochore | 0.4 | (0.2) | 0.9 | (0.2) | 1.3 | (0.3) | 20.5 | |
| Endozoochore | 2.9 | (0.5) | 2.9 | (0.5) | 3.4 | (0.5) | 20.0 | |
| Barochore | 3.9 | (0.7) | 6.1 | (0.9) | 6.5 | (0.6) | 15.2 | |
| Anemochore | 2.8 | (0.4) | 3.6 | (0.2) | 4.8 | (0.9) | 13.1 | |
| Mobile | 4.4 | (0.6) | 4.9 | (0.6) | 4.9 | (0.4) | 8.5 | |
| Fire response | | | *Pseudo-F* = 2.1, *p* = 0.0353 | | | | | |
| | a | | ab | | b | | CTRF v WF | |
| Obligate seeder (S) | 4.3 | (0.4) | 5.8 | (0.5) | 6.9 | (0.9) | 26.8 | |
| Obligate resprouter (R) | 6.0 | (0.9) | 7.4 | (0.9) | 8.2 | (0.7) | 26.4 | |
| Facultative seeder (SR) | 1.4 | (0.2) | 1.2 | (0.2) | 1.5 | (0.3) | 22.2 | |
| Weak seeder (Sr) | 1.5 | (0.3) | 2.1 | (0.3) | 2.4 | (0.2) | 16.8 | |
| Weak resprouter (Rs) | 3.4 | (0.3) | 3.6 | (0.4) | 4.2 | (0.4) | 7.8 | |
| Seral stage | | | *Pseudo-F* = 7.2, *p* = 0.0003 | | | | | |
| | a | | b | | b | | CTRF v WF | CTRF v MF |
| Early | 3.1 | (1.3) | 4.2 | (1.0) | 6.7 | (1.5) | 53.1 | 48.0 |
| Rainforest | 4.9 | (1.2) | 3.3 | (0.9) | 2.3 | (0.6) | 24.3 | 29.8 |
| Generalist | 8.3 | (1.4) | 12.2 | (1.0) | 13.6 | (0.9) | 22.7 | 22.1 |
| Origin | | | *Pseudo-F* = 7.0, *p* = 0.0036 | | | | | |
| Native | 16.3a | (1.5) | 19.5b | (0.7) | 22.0b | (1.1) | | |
| | | | *Pseudo-F* = 1.9, *p* = 0.1649 | | | | | |
| Introduced | 0.8 | (0.3) | 1.1 | (0.3) | 1.5 | (0.5) | | |
| | | | *Pseudo-F* = 8.7, *p* = 0.0015 | | | | | |
| All species | 18.2a | (1.9) | 22.1b | (0.9) | 25.3b | (1.6) | | |

Mean pairwise similarity in composition among forest types was greatest between MF and WF (43%) with reduced similarity between CTRF and WF (34%); and CTRF and MF (35%) and PERMANOVA indicated a significant (*Pseudo-F* = 2.6, *p* = 0.0024) turnover among forest types that is reflected in the clear separation of sites in the NMDS ordination (Figure 4). SIMPER indicated that many species each made small contributions to differences among forest types, including *Clematis aristata* and *Cassinia aculeata* that were more abundant in WF and MF relative to CTRF (Figure 4, Tables S8 and S10).

## 4. Discussion

Understanding how CTRF seed banks respond to fire-related germination cues is integral to understanding how they will respond to increasing fire severity and frequency under climate change. These changes are likely to affect the regeneration of floristically diverse ecosystems and examining these differences is important for the preservation of CTRF. Four questions were asked about germination cues and diversity of species in CTRF, mixed forest and wet forest soil seed banks.

### 4.1. Does the Soil Seed Store Respond to Fire-Related Germination Cues and Do These Cues Differ among the Three Forest Types?

In support of our hypotheses, fire-related germination cues were consistent among species shared across forest types with requirement for high-heat concentrated in ant-dispersed species with hard seed coats. There were however no differences in species diversity, abundance, or rates of seedling emergence in response to different germination treatments. Contrary to our hypothesis, increase in temperature did not lead to a decline in rainforest-associated species. However, the richness and abundance of endozoochores declined with increase in temperature, with the greatest store of these species located in CTRF, although in low densities.

The lack of strong fire-related germination cues across theses three forest types is likely due to their association with 'wet' or low-flammable forests [37,38,79]. Although these forests may experience higher fire intensity and frequency in the future, they have not historically been subject to these conditions [80]. CTRF, mixed forest, and wet forest species may be more responsive to non-fire disturbances for promoting the germination of the soil seed bank. For example, the superb lyrebird is thought to be a critical ecosystem engineer in these forest types due to the large amounts of soil and litter that they turn over, resulting in a physical disturbance to the soil seed store and exposure of seeds to greater light environments. Superb lyrebirds spend most of their time in these forest types and can decrease compaction by 37% and decreased litter depth threefold so may play a more critical part in soil seed bank germination in these wet ecosystems than fire does [81]. Moreover, other studies have demonstrated that the act of physical disturbance can promote germination in the absence of these field-related cues [55].

Specific to these wet forests, Asthon and Chinner [82] documented positive effects of soil scarification on emergence of woody understorey species. Similarly, along an ecotone supporting dry sclerophyll, wet sclerophyll, and rainforests in south-east Australia, Campbell et al. [37] demonstrated soil disturbance increased recruitment and survival of rainforest species above that in burnt sites, while shrubs associated with the more flammable dry sclerophyll forests recruited and survived in both burnt and unburnt forest. Gap-phase recruitment from seed stored in soil seed banks or in fleshy fruits may also play an important role for species lacking post-fire seedling recruitment [83]. For example, *Nothofagus* and *Atherosperma* have greater germination in tree fall gaps relative to undisturbed forest, with seedlings often concentrated on coarse woody debris [84–87]. Further, the tree fall gaps are often filled with dense tree ferns with the trunks providing a germination niche for many species, including *Nothofagus* and *Atherosperma* [84,88].

The positive response of hard-seeded, ant-dispersed species, and particularly acacias to high heat is consistent with previous work and reflects the long-lived highly dormant soil seed banks in these species and recruitment in response to infrequent landscape-scale

disturbance. This is typical of legumes in Mediterranean-type climates, but also of those in mesic forests [15,64]. Moreover, seed burial by ants may act to protect physically dormant seed from heat induced mortality associated with the more extreme temperatures at the soil surface. Seed burial may also increase the post-fire residual soil seed bank given temperature decreases rapidly with soil depth so that temperatures may fall short of those required to break physical dormancy [89].

### 4.2. Does the Application of Karrikinolide Increase Germinant Diversity and Abundance and Is This Dependent on the Application of Heat?

Karrakinolide did not result in an increase in richness or abundance of soil seed bank germinants above vermiculite-infused smoke treatment (S) or control (C) and this was consistent across all heat treatments. In contrast to our hypothesis, rainforest-associated species did not demonstrate a stronger response to K than mixed forest or wet forest species. Similarly, S was not more effective than C in promoting germination, irrespective of heat treatment.

The lack of response to smoke cues, and in this instance, both K and S, is consistent with decreased incidences of smoke-induced germination in south-eastern Australian flora relative to other fire-prone areas, and of lower incidences of smoke-induced germination in wetter areas relative to drier habitats [46]. In contrast to Carthey et al. [46] we found no indication of response of obligate seeders to smoke cues in the absence of heat. These findings are however consistent with our earlier work in wet forests [15] and consistency across the K and S treatments suggests this lack of response is not related to the method of smoke application. Aerosol smoke, like K, is often reported to be more effective than other smoke products [46,55] but remains to be tested in these forest types. The lack of K effect on seed bank germination (heat independent or otherwise) suggests increased light availability and/or release from physical dormancy associated with soil disturbance by fauna or tree fall is more important for seed germination than non-fire related pathways (e.g., microbial decomposition of organic matter, [52]) for karrikinolide production.

### 4.3. Does the Abundance and Diversity of Soil Seed Banks Differ among Forest Types?

In part support of our hypothesis, wet forest, but not mixed forest supported a larger and more diverse soil seed bank than rainforest, including greater richness and abundance of early successional species and consistent with Egler's [14] initial floristics model of succession reported for these wet forests [13,15]. In contrast, rainforest soil seed banks supported a greater richness and abundance of rainforest associated species than mixed forest and wet forest, although densities were low. The most abundant rainforest associated species was *Acacia melanoxylon* and this species made the greatest contribution to dissimilarity between CTR and both mixed forest and WF. This reflects the broader accumulation of hard-seeded, ant-dispersed species belowground and requirement of fire cues for germination [15,89].

*Zieria arborescens* and *Pomaderris aspera* made the greatest contributions to dissimilarity of WF with MF and CTRF and there was a general trend for a decline in abundance across the WF to CTRF ecotone. These species are both fire sensitive obligate seeders and rely on regeneration from the soil seed store following fire. The hard-seeded, ant-dispersed *P. aspera* will regenerate on mass following fire, with gradual decline with increasing time since fire, although resident trees can persist under low light conditions and provide to the accumulation of soil stored seed [90,91]. *Zieria arborescens* is similarly shade tolerant and provides an ongoing contribution to the soil seed store during the inter-fire interval [92]. The species also demonstrates strong regeneration following fire [93] although seed do not respond to fire cues (this study, [15]). For this species, persistent seed stores may be related to depth in the soil profile with Floyd [92] reporting viable seed up to depths of 12 cm. White and Vesk [93] suggest soil seed banks of *Zieria arborescences* may be unable to survive high intensity wildfire in the absence of an intact insulating litter layer following forest harvesting. Our work has however demonstrated that this species can survive high

heat treatments of 90 °C (this study, [15]) with soil surface temperatures of 900 °C during planned burns quickly declining to less than 100 °C over the first 5 cm of the soil profile (S. Kasel, unpublished data).

*4.4. Does the Soil Seed Bank Reflect Species Diversity in the Extant Vegetation and Does This Differ among the Three Forest Types?*

The soil seed bank did not reflect species diversity in the extant vegetation, and this was consistent among forest types. Of the 108 total recorded species, only 29 (27%) were shared between the above ground vegetation and the soil seed bank. Consistent with our hypothesis, rainforest-associated species were concentrated in extant vegetation and poorly represented in the soil seed bank.

Of the 17 rainforest affiliated species, eight were ferns and none of these were recorded in the soil seed bank. The two ferns present in the soil seed bank (*Polystichum proliferum* and *Histiopteris incisa*) are not considered rainforest indicator species [15,94]. Of the other nine rainforest associated species only *Acacia melanoxylon* (221 germinants), *Leptostigma breviflorum* (2) and *Nothofagus cunninghamii* (1) were recorded in the soil seed bank. Three species (*Acacia frigescens*, *Dianella tasmanica*, *Persoonia arborea*) were recorded in the soil seed banks of previous work albeit in very low densities [15], suggesting seed was either not viable or not present, rather than viable and ungerminated due to lack of suitable germination cues.

Increased abundance of *Acacia melanoxylon* in the soil seed bank, is consistent with temporal accumulation of a long-term persistent soil seed store in hard-seeded acacias [89]. Regeneration of *Acacia melanoxylon* is however impacted by competition as it requires high light irradiance [95] and therefore dependent on triggers that alleviate physical dormancy and provide for increased light availability. Only one (*Leptostigma breviflorum*) of five fleshy-fruited rainforest associated species germinated from the soil seed bank. This is consistent with the premise that fleshy-fruited plants do not form a persistent soil seed bank and that their germination is not tied to fire, especially in non-fireprone habitats such as rainforests [40,96]. Despite its presence in extant vegetation, *Pittosporum bicolor* was absent from the soil seed bank, consistent with previous work on seed rain and seed bank germination trials (Kasel unpublished data, [15]). This frugivorous species may require additional germination cues that mimic digestion to remove an inhibiting sticky seed coating, consistent with the use of the artificial enzyme pancreatin in faster germination of this species in a commercial setting [97]. *Pittosporum bicolor* is a common epiphyte on other species including tree ferns (*Dicksonia antarctica*), *Nothofagus cunninghamii* and *Eucalyptus regnans*, that may reflect seed deposition by birds at these sites [88].

The lack of representation of rainforest overstorey (*Nothofagus cunninghamii*, *Atherosperma moschatum*) in the soil seed bank is consistent with previous work [15,98,99] and suggests regeneration from seed relies on dispersal from surrounding stands [35,41], with both species also capable of resprouting from surviving basal stems [28,29,100]. *Nothofagus cunninghamii* regeneration from seed is typically restricted to 20 m from the source with mast seedfall approximately every three years [101]. The wind-dispersed seed of *Atherosperma moschatum* can disperse over 150 m [100,101]. Disturbed or burnt soil promotes the germination of both *N. cunninghamii* and *A. moschatum* [95] with coarse woody debris and trunks of tree ferns also providing protected microsites for germination [84–87] that may be particularly important for *A. moschatum* that is highly palatable to browsing mammals [101,102].

## 5. Conclusions

Fire germination cues do not play a strong role in the germination of soil stored seed banks in CTRFs of south-eastern Australia. For these threatened ecosystems, other non-fire disturbances such as treefalls and soil movement by fauna may be more important for germination. Soil seed bank response to soil disturbance would most likely be associated with mechanical seed abrasion and/or exposure to increased light availability than to non-

fire related production of smoke products given the lack of response to karrikinolide. The predominance of fire-cued persistent soil seed stores of acacias across all three forest types lends further support to the potential for replacement of obligate seeding wet-sclerophyll eucalypt forests with acacia shrubland with increases in fire frequency. Further, this study has demonstrated that this potential extends into CTRF given rainforest overstorey species lack canopy and soil seed stores, with seed regeneration reliant on dispersal from fire refugia. However, local topographic controls important for these rainforest fire refugia are becoming increasingly overridden by severe fire weather conditions as fire severity increases with climate change [103,104]. These rainforest canopy species may become increasingly reliant on resprouting for post-fire regeneration, although the number of fire events and associated resprouting cycles these trees can tolerate is unclear [10,29]. Further work examining short-term persistent seed banks in the litter layer, the geographical extent and composition of seed rain inputs, and germination niches for rainforest dependent species, and in particular for understorey species, will enhance our understanding of successional dynamics in this wet forest to mixed forest to CTRF ecotone of south-eastern Australia.

**Supplementary Materials:** The following supporting information can be downloaded at: https://www.mdpi.com/article/10.3390/fire6040138/s1. Figure S1: Location of study sites selected from previous work conducted by Fedrigo et al. [26]. Figure S2: Relative proportion (%) of germinable seeds within life form, dispersal mode, fire response, and seral stage functional types. Figure S3: Cumulative number of germinants within each of the 10 soil seed bank treatments. Figure S4: Soil seed bank treatment (C, control; S, smoke only; K, Karrikinolide; VLH, Very Low Heat; LH, Low Heat; HH, High Heat) and forest type (CTRF, cool temperate rainforest; MF, mixed forest; WF, wet forest) effects on richness (circles) and germinant density (bars) (mean per site, standard error) of species according to life form, dispersal mode, fire response and seral stage. Figure S5: NMDS ordination of abundance data for trait associations in the germinable soil seed bank across three forest types (CTRF, cool temperate rainforest; MF, mixed forest; WF, wet forest). Table S1: Spatial, topographic, edaphic, and structural variables across the three forest types sampled in the Central Highlands region. Variables were measured during the initial site establishment as reported in Fedrigo et al. [26], methods follow [105–111]. Table S2: Total number of seedlings per treatment and per forest type in soils from cool temperate rainforest, wet forest, and wet forest in the Central Highlands of south-eastern Australia. Trait values sourced from [15,70,112–114]. Table S3: Treatment × Forest Type effects on soil seed bank: richness (a), abundance (b), diversity (c), and composition (d). Table S4: SIMPER analysis of significant forest type effects on species composition of the soil seed bank. Table S5 Significance of Treatment (Tr) and Forest Type (FT) effects on abundance and richness of functional attributes within the soil seed bank. Table S6: Treatment × Forest Type effects on functional trait associations in the soil seed bank. Table S7: SIMPER analysis of forest type effects on trait associations. Table S8: Vegetation Pool x Forest Type effects on species composition. Table S9: SIMPER analysis of significant vegetation pool effects on species composition (see Table S8). Table S10: SIMPER analysis of significant forest type effects on species composition (see Table S8).

**Author Contributions:** Conceptualization, S.K.; methodology, S.K. and S.Y.; software, S.K.; validation, S.K. and S.Y.; formal analysis, S.K.; investigation, S.Y.; resources, S.K.; data curation, S.K.; writing—original draft preparation, S.Y.; writing—review and editing, S.K. and S.Y.; visualization, S.K.; supervision, S.K.; project administration, S.K.; funding acquisition, S.K. All authors have read and agreed to the published version of the manuscript.

**Funding:** This research was funded by the Victorian Department of Energy, Environment and Climate Action (DEECA), under the Integrated Forest Ecosystem Research Program (IFER) grant number 301103.

**Institutional Review Board Statement:** Not applicable.

**Informed Consent Statement:** Not applicable.

**Data Availability Statement:** The data presented in this study are available on request from the corresponding author.

**Acknowledgments:** We acknowledge the traditional owners of the lands on which this research was conducted, the Wurundjeri, Taungurung and Gunaikurnai people. We pay our respects to their elders past and present. This work was undertaken with Approval by the Victorian Department of Energy, Environment and Climate Action (DEECA) under Research Permit Number 10009575. We would like to thank Sascha Andrusiak and Rowan Berry for support with glasshouse facilities; Ben Smith and David Lockwood for contribution to data collection; and Lauren T. Bennett for earlier comments on the manuscript. We thank two reviewers for comments that improved the manuscript.

**Conflicts of Interest:** The authors declare no conflict of interest.

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
