# Peer review of "Do Fire Cues Enhance Germination of Soil Seed Stores across an Ecotone of Wet Eucalypt Forest to Cool Temperate Rainforest in the Central Highlands of South-Eastern Australia?"

_fire, doi:10.3390/fire6040138_

Round 1
Reviewer 1 Report
Review of Younis and Kasel
I have reviewed the manuscript by Younis and Kasel titled ‘Do fire cues enhance germination of soil seed stores across an 2 ecotone of wet eucalypt forest to cool temperate rainforest in 3 the Central Highlands of south-eastern Australia?’.
The manuscript presents results of a study of soil seed bank germination in response to simulated fire effects (heat and smoke treatments), with these effects compared across three vegetation types (wet eucalypt forest, mixed forest, and temperate rainforest) from the Central Highlands of Victoria, Australia.
The main finding was that vegetation type was a much stronger driver of floristics in the germinated seed bank than were the simulated fire germination cues. Smoke treatments had limited, if any, influence on floristics in the germinated seed bank. These findings are not dissimilar to prior findings for these vegetation types, and are, in the author’s words, consistent with disturbance types other than fire having important roles in plant community dynamics in these systems.
Overall, I think this study makes a nice contribution to our understanding of seed bank germination in these forest types, and the degree to which fire and specific fire cues are involved (or not) with this process. I like the functional trait approach, and I think the interpretation of results are valid and within the scope of the data. I don’t have any major concerns with the way the study was conducted. I only have some minor comments / suggestions that I hope can enhance an already strong manuscript. I do think it would have been good to see what the abundances of seeds were present in ungerminated soil, but I don’t think this is strictly necessary to test the hypotheses.
My specific comments are below:
Abstract: I think it would be good to give an idea of replication (n = 5 if I understand correctly?) somewhere around line 13-15 when outlining the study design. I also think a concise summary of the primary conclusion at the end of the abstract would be valuable.
I wondered if wet eucalypt forest could be used throughout, rather than simply ‘wet forest’? I’m not sure that moisture regime is best way of differentiating these three vegetation types and I found it a bit confusing to read about ‘wet forest’ versus ‘rainforest’.
Similarly, I wonder if the initialisms ‘K’ and ‘S’ are really necessary (for the treatments)? I don’t think using the whole word makes things clunky and it would avoid any potential confusion for potassium and sulfur treatments.
Line 44: The sentence beginning here could use some finessing – it reads like mature forest is the predominant disturbance and ends up a bit confusing.
Lines 53-55: Perhaps this point should be constrained to E. regnans forest? Partly because E. regnans has already been identified as a focal species in the preceding paragraph.
Line 64-81: see also the work by Crockett, Mackey, and Ash ‘The Role of Fire in Governing the Relative Distribution of Rainforest and Sclerophyll Forest: the Effects of Rainforest Vegetation on Fire Spread.’
Lin 103: Perhaps ‘induce germination of’ would be clearer?
Line 103-105: Could you clarify this a bit more? I think it means that the richness and abundance of seedlings germinating from the soil seedbank are increased by smoke?
Line 138: Part iii of Q1 could be written a little more clearly. Greater response to smoke at no or low heat?
Line 225: PERMANOVA is missing the ‘V’.
Lines 293-300: It’s a bit unclear what these percentages refer to, as they add up to >100%.
Figure 1b : the letters over forest types suggest significant difference between CTRF and WF for number of germinants, but the means and error bars look very similar / overlap substantially. Are the authors sure the post hoc test results are correct on this panel? Perhaps a result of the transformation applied…
Figure 2: I find the design of these hybrid plots difficult to interpret. It seems like negative error bars are provided for bars but positive errors provided for circles? But panel d deviates from this…(because error > mean?). Normally both positive and negative errors would be shown. Then having the post-hoc letter displays at the bottom of bars, positioned inconsistently, adds further confusion (for me at least). I think the information would be easier to interpret if the variables (no. of germinants and species richness) were simply plotted separately, with positive and negative errors in all cases. There’s plenty of space to achieve this within one figure.
Reviewer 2 Report
A brief summary
Extreme forest fires are becoming more common due to climate change. Even in forests where wildfires are an integral part of their life cycle (i.e., Australian eucalyptus forests), the predicted anomalous increase in the frequency of severe fires in the future could significantly affect biodiversity and post-fire successions. However, the number of fire events and associated regenerating cycles these tree stands can tolerate are poorly understood and unclear. The manuscript represented a multivariate study of fire-related germination cues of soil seed banks along a continuum of transition from wet forest, to cool temperate mixed forest, and cool temperate rainforest.
General concept comments
The topic of the study is relevant and fully reflect the essence of the study. The manuscript is presented in a well-structured manner and includes 5 common sections.
Introduction. This chapter represented the justification for the choice of research topic and key research gaps existing in this field as well as hypotheses considered in the study.
Materials and Methods include the description of all used methods for investigation of described study sites. It should be stated that experimental design was appropriate to test the hypothesis.
Results are perfectly illustrated and statistically proven. Provided tables and figures show the data properly, and they are easy to interpret and understand.
Discussion structured in accordance with the considerув hypotheses and supplemented by the data available in the literature.
Conclusions provides key findings from the study that are consistent with the main goal. Аdditionally, it is recommended to emphasize the future prospects of this study since they undoubtedly exist and are important for understanding the successional dynamics of forest ecotone in the Central Highlands of south-eastern Australia.
Reference list contains publications predominantly before 2017. Only 22 from 102 references are after 2018. Authors should enlarge the number of cited recent articles (within the last 5 years).
Supplementary materials are wide and perfectly complement the main content of the manuscript. But Figure S1 should be transferred in subsection 2.1 to illustrate better study sites in the manuscript body
Specific comments
Line 29 Check the spelling “karrikinolide”
Line 201 Authors indicated that Table 1 should contain not only the temperature conditions of soil seed bank treatment but also their duration. But this information is absent in the Table. Authors should add this information for each treatment type.
Line 253 Write the formula of Shannon’s diversity index in a separate line according to journal requirements
Line 253, 258 and 265 Check if the transformation formulas are correct. They are different: log10(x+1) and log10x + 1
Line 255 Check the spelling correctness of PERMANOVA
Line 267 Write “Kolmogorov-Smirnov test” in singular
